# Breathing Freely: Self-supervised Liver T1rho Mapping from A Single T1rho-weighted Image

**Chaoxing Huang**[1]               CHAOXING.HUANG@LINK.CUHK.EDU.HK
**Yurui Qian**[1]                  YURUI.QIAN@LINK.CUHK.EDU.HK
**Jian Hou**[1]                     HOUJIAN@LINK.CUHK.EDU.HK
**Baiyan Jiang**[1,3]                 BAIYAN.JIANG@GMAIL.COM
**Queenie Chan**[4]                QUEENIE.CHAN@PHILIPS.COM
**Vincent Wong**[2]                WONGV@CUHK.EDU.HK
**Winne Chu**[1]                   WINNIECHU@CUHK.EDU.HK
**Weitian Chen**[1]                WTCHEN@CUHK.EDU.HK

[1] *Department of Imaging and Interventional Radiology, The Chinese University of Hong Kong*

[2] *Department of Medicine and Therapeutics, The Chinese University of Hong Kong*

[3] *Illuminatio Medical Technology Limited, Hong Kong SAR, China*

[4] *Philips Healthcare, Hong Kong SAR, China*

## Abstract

Quantitative $T1rho$ imaging is a promising technique for assessment of chronic liver disease. The standard approach requires acquisition of multiple $T1rho$-weighted images of the liver to quantify $T1rho$ relaxation time. The quantification accuracy can be affected by respiratory motion if the subjects cannot hold the breath during the scan. To tackle this problem, we propose a self-supervised mapping method by taking only one $T1rho$-weighted image to do the mapping. Our method takes into account of signal scale variations in MR scan when performing $T1rho$ quantification. Preliminary experimental results show that our method can achieve better mapping performance than the traditional fitting method, particularly in free-breathing scenarios.

**Keywords:** magnetic resonance, $T1rho$ mapping, liver-tissue, self-supervised learning

## 1. Introduction

The ability of indicating liver inflammation noninvasively and the sensitivity to the change of macromolecular contents in tissue makes quantitative T1rho imaging a promising technique for assessment of chronic liver disease. To quantify T1rho, multiple T1rho-weighted images are acquired with various time-of-spin-lock (TSL). The T1rho relaxation time is calculated by fitting these images to a known relaxation model. Misalignment between T1rho-weighted images can lead to fitting errors. To avoid image misalignment, T1rho-weighted images of the liver can be acquired using either free-breathing techniques or breath-hold. The free-breathing method results in significantly prolonged scan time and it may suffer from residual motion. The breath-hold approach typically requires a breath hold of 8 to 10 seconds to acquire 4 to 5 T1rho-weighted images of a single slice (Wáng et al., 2018b). Even though breath-hold is expected to provide better motion compensation, it creates extra burden to the workflow of MRI scan. For some patients, they may have difficulties to hold breath sufficiently long. In this work, we report our investigation of the feasibility of mapping $T1rho$

value from only one $T1rho$-weighted image by using a self-supervised learning approach to tackle the aforementioned problems.

The adaptation of deep learning method in quantitative MRI has recently gained interests, and most of the works focused on T1 mapping and T2 mapping. A comprehensive review can be seen in (Feng et al., 2020). Liu et al.(Liu et al., 2019) and Jeelani et al. (Jeelani et al., 2020) treat the fitted quantification map from fully sampled k-space images as the ground-truth, and train a network for mapping multiple under-sampled images to the fully-sampled T2 and T1 map respectively. Similarly, a MR fingerprint (MRF) quantification based work (Fang et al., 2019) treats the dictionary-matching result as the ground-truth of quantification. Those supervised learning methods requires extra fitting process and a considerable amount of collected data. In addition, the pixel wise fitted "ground-truth" may not be the gold-standard due to noise and artifacts (Chen, 2015), and the supervised learning method is lack of physics intuition. Recently, Liu et al. (Liu et al., 2021) and Varadarajan et al. (Varadarajan et al., 2021) used the physics attribute of T1 and T2 to do the data consistent self-supervised learning. Directly applying them to $T_1rho$ mapping may overfit on physics parameter intrinsically (proton density, hardware and platform scaling, etc, ) since the signal scale variation problem (Filo and Mezer, 2018) is not addressed. Similarly, Kang et al. (Kang et al., 2021) incorporated the analytical MTC-MRF equation into the self-supervised learning. Wu et al. (Wu et al., 2021) demonstrates a supervised single scan mapping method . Li et al. (Li et al., 2020) first proposed a supervised method for joint mapping of $T1rho$ and T2 in cartilage using multiple $T1rho$-weighted and T2-weighted images. To our best knowledge, none of those previous works focus on facilitating liver $T1rho$ mapping with deep learning, and fully address signal scale variation problems. In this work, we propose a self-supervised mapping method by taking only one $T1rho$-weighted image to facilitate the demands of liver $T1rho$ mapping under free breathing and different signal scales.

Our contributions are as follows: **1)**Our work is the first tackling the problems of deep learning based liver $T1rho$ mapping. **2)**We propose a self-supervised mapping method using one single $T1rho$-weighted image and avoids the challenging tasks of acquiring the ground truth. **3)**We propose a two-branch joint learning mechanism to reduce the sensitivity of $T1rho$ prediction to signal scale variations. **4)**We demonstrates the reliability of our method when respiratory motion exits, and shows the potentials of this method for free breathing acquisitions.

## 2. Method

### 2.1. $T1rho$ relaxation model

$T1rho$ refers to the spin-lattice relaxation time in the rotating frame. For on-resonance spin-lock, the corresponding signal model is described as (Yuan and Wang, 2016):

$$I(T_s) = S_0 \exp(-\frac{T_s}{T1rho}) \tag{1}$$

where $T_s$ is the duration of the spin-lock RF pulse (TSL) and $S_0$ is an unknown constant independent from TSL. The contrast information in the image reflects the $T1rho$ differences between tissues. Traditionally, multiple $T1rho$ weighted images at various TSLs are

acquired and the images are fitted to model described by Equation (1) to calculate $S_0$ and $T1rho$ within the region of interests (ROI) or at every pixels. The information of proton density, T1 and T2 relaxation time, parameters of pulse sequence settings, hardware and vendors' reconstruction system can all affect $S_0$. Uniformly scaling is common for collected $T1rho$-weighed images and so does for $S_0$. Note the values of $T1rho$ are inherent tissue properties and should not change at different values of $S_0$. In this work, we will use the physics model of $T1rho$ to form our self-supervised learning pipeline.

## 2.2. Relaxation-informed self-supervision

### 2.2.1. RELAXATION MODEL BASED LOSS FUNCTION

Our goal is to fit a model that takes one single $T1rho$ weighted image as input, and results in the output $S_0$ and $T1rho$ simultaneously. Let us denote a group of $T1rho$ weighted images of the same slice with $n$ different time of spin-lock as $G = \{I(T_s^i)|i = 1, 2, \ldots, n\}$ in the training set, one would think of using the $S_0^i$ and $T1rho^i$ being obtained from $I(T_s^i)$ to synthesize another $T1rho$ weighted image $I(T_s^j)$ in group $G$ using Equation (1). The loss function can therefore be written as:

$$L_1 = \sum_{g=1}^{N} \sum_{i=1}^{n} \sum_{j=1}^{n} \left\| S_0^i \exp(-\frac{T_s^j}{T1rho^i}) - I(T_s^j) \right\| \tag{2}$$

where $N$ is the number of slices in the training set. This loss function is to ensure the obtained $S_0$ and $T1rho$ can be used to synthesize other $T1rho$ weighted images from the same slice, including the input itself .

### 2.2.2. PIPELINE

We adopt a two-branch joint training mechanism with physics intuition to fit $S_0$ and $T_1rho$. Specifically, the input of the $S_0$ branch is the $T1rho$ weighted image in raw form, since it contains the information of signal scaling of the $T1rho$ weighted image. Different images have different range of the signal. Normalizing every input $T1rho$ weighted image to the same range (e.g., [0,1]), on the other hand, creates only contrast information without uniform scale information encoded in $S_0$. The network of $S_0$ branch is an UNet (Ronneberger et al., 2015) architecture from (Buda et al., 2019). A one-channel convolution is added at the end as the output layer. As the appearance of $S_0$ is similar to the input image, a mask is first learnt and then it is multiplied by the input image to obtain $S_0$. We find it easier for the model to converge in this way since the appearance information of the input $T1rho$ weighted image is utilised.

As for the $T1rho$ branch, we use the same network architecture with the previous branch, but with different network parameters. Directly applying the raw image to fit $T1rho$ would intrinsically overfit on those scaling-related parameters. Since the $T1rho$ value is an inherent tissue property and should not be affected by those parameters, we use the contrast information as the input. The input of this branch is the $T1rho$ weighted image normalized to the range of $[0, 1]$. The output of the last layer is directly regarded as the $T1rho$ map without any mask multiplication. Once we get $S_0$ map and the $T1rho$ map, we can plug

them into the loss function in Equation (2) for training. The whole pipeline is shown in Figure 1.

During testing, the $S_0$ branch is discarded and we will only use the branch of $T1rho$ to infer the $T1rho$ value. No matter what scale the input signal is at, the input of this branch is always the same within the range from 0 to 1.

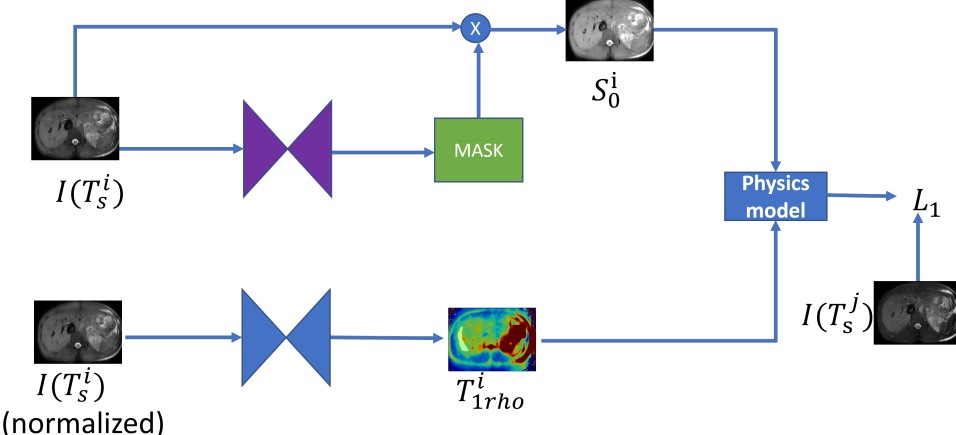

Figure 1: Training pipeline of the model

## 2.3. Evaluation Metric

The reliability of pixel-wise $T1rho$ analysis can be affected by noise (Chen, 2015), and the mean value analysis within the ROIs is used to reduce noise influence. Inspired by the ROI-based evaluation method used in the previous deep learning based quantification works (Jeelani et al., 2020; Sveinsson et al., 2021; Qiu et al., 2021), we calculated $T1rho$ map using 4 $T1rho$-weighted images by the traditional fitting method, and use the mean T1rho value in an ROI at the right lobe of the liver as the reference value. The average of all the prediction-wise absolute error between the mean of the output $T1rho$ map in the ROI and the reference value is the evaluation metric, which is shown in the equation below where $P$ stands for the total number of predictions. The ROI example is indicated in Figure 2. Note all the predictions of $T1rho$ are calculated as the mean value in the ROI in this work.

$$Error = \frac{1}{P}\sum_{p=1}^{P}\left\|\overline{T1rho^p(u,v)} - \overline{\widehat{T1rho^p(u,v)}}\right\|_{(u,v)\in ROI_p} \tag{3}$$

## 2.4. Data Acquisition

All of the scans were conducted using a 3.0 T MRI scanner (Philips Achieva TX, Philips Healthcare, Best, Netherlands). A 32-channel cardiac coil (Invivo Corp, Gainesville, USA) was used as the receiver and the body coil was used as the RF transmitter. The $T1rho$ imaging data was acquired using the pulse sequence described in (Chen et al., 2016) with

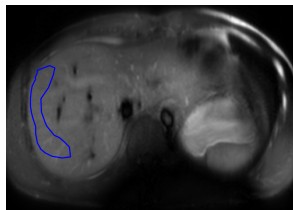

Figure 2: The ROI example during testing

those parameters: time of spin-lock (TSL) = [0, 10, 30, 50] ms, frequency of spin-lock (FSL) = 300 Hz, imaging resolution = 2 × 2 mm, slice thickness = 7 mm, repetition time (TR) = 2000 ms, echo time (TE) = 17 ms. Three axial section slices of the liver were acquired. The fat and blood signals were suppressed through spectral pre-saturation with inversion recovery (SPIR) and double inversion recovery (DIR), respectively. A shimming box was placed on the right lobe of the liver to reduce the susceptibility effect. The subjects were instructed to hold the breath during scan.

## 3. Experiments

### 3.1. Dataset

The in vivo studies were approved by the institutional review board. We first retrospectively fetched the data from 20 healthy subjects using the above acquisition protocol. Among those 20 healthy subjects, two males and two females were randomly selected as the healthy test-set (HT), while one subject was randomly chosen for validation. The rest 15 subjects were used for training. We also retrospectively included 41 patients with alcoholic fatty liver disease. The $T1rho$ data of those patients were acquired using the same aforementioned protocol. One male patient with apparent motion was discarded, and the rest of the patients were used for testing only to verify the generality of our model to clinical data. We refer them as clinical test-set (CT). To examine the performance of our model on data with respiratory motion, we additionally collected the data from two healthy males and three healthy females under free breathing conditions. Along with the previous clinical subject with apparent motion, these subjects formed the test-set under free breathing conditions (FBT).

### 3.2. Implementation details

All the images were resized to 256 × 184. We applied data augmentation to every images in the training set, with slight rotations and translations. We discarded the scan with $Ts = 0$ since they did not have $T1rho$ weighting.There were eventually more than 11000 numbers of training pairs for the self-supervised learning. The test ROI of every slice was manually drawn on the $T1rho$ weighted images with TSL = 0 and the drawing was carried out before any kind of fitting and testing to ensure the fairness of the evaluation. When drawing the test ROI, we focused on the liver parenchyma by avoiding the area with large vessels and artifacts, following the principles in those previous works (Wáng et al., 2018b; Allkemper

et al., 2014; Chen et al., 2018). The batch size was 4, and the learning rate was 1e-4. Adam (Kingma and Ba, 2014) was used as the optimizer. The deep learning framework for the implementation was Pytorch 1.9 (Paszke et al., 2019) and the experiments were conducted on one GTX 1080ti GPU for three hours.

### 3.3. Comparison study

We first conducted the comparison study and tested the model on healthy subjects (HT) and patients (CT). Since we want to reduce the time for breath-holding during scan, we set the performance of non-linear least square fitting method using two $T1rho$ weighted images from two TSLs (2-TSL) as the benchmark. We also trained two 2-branch models with the same architecture but with raw images or normalized images as the input for both branches. The results were shown in Table 1. Our two-branch model with normalized input achieved a better performance than the 2-TSL fitting model on both HT and CT data sets with less input information (with only one T1rho-weighted image). The 2-branch model also performed better than the 2-branch model that took the raw images as input. The 2-branch (all normalized) produced inferior results since it violated the physics that it is impossible to map $S_0$ with scaling information from purely contrast information. Besides, the input is scaled under different signal scaling factors to mimic the uniform scaling in reality. The 2-branch model (raw signal input) over-fitted on the signal scales similar to the training data, while signal scale variations did not affect the performance of the proposed model. The mean prediction for every subjects is computed and a pair t-test were carried out to test if the subject-wise mean $T1rho$ is significantly different from the reference. The $p$ **value** of the benchmark and proposed method was **0.008** and **0.15**, respectively, which demonstrated the superiority of our method. Since the two branch model had predictions of three different TSLs for each slice, we also computed its intraclass correlation coefficient (ICC) on the patient dataset to examine the repeatability. The **ICC** was **0.81**, indicating a decent repeatability.

### 3.4. Free Breathing Scenario

We further compared the performance of our model and multi TSL fitting methods on data sets acquired with free breathing. Since a perfect slice-wise ground-truth was not available, the evaluation were in two ways: **1)** We calculated the mean prediction value per person to inspect if the value was within the typical range of liver $T1rho$ values qualitatively. Note the typical T1rho value of healthy liver measured using the aforementioned pulse sequence is approximately within 30ms to 50ms (Wáng et al., 2018a). **2)** Breath-hold data of the five healthy volunteers was acquired using the same FOV and shim box positional planning as the free breathing scenario and a paired hypothesis testing was carried out. Each scan with a specific positional planning is refereed as one session. Scan sessions from both scenarios with the same planning were therefore paired. Note the slice positions of the sessions using the same planning from these two scenarios are not strictly the same due to out of plane motion. We still considered those two sessions as one pair. We refer readers to Appendix A for further explanation. A session-wise Wilcoxon signed rank test (Wilcoxon, 1992) was carried out to test the null hypothesis that the session-wise mean prediction had the same distribution as the 4-TSL fitted reference value in breath-hold scenario.

Table 1: Performance of the model under different settings and that of 2-TSL case. The ROI mean value in 4-tsl case is regarded as the reference value. The value inside the bracket is the standard deviation.

| Test-set | **HT** | **CT** | *Scale factor* |
|---|---|---|---|
| Models | Error (ms) | Error (ms) | |
| 2-TSL | 5.39(6.09) | 4.03(5.81) | |
| 2-branch (raw input) | 3.59(1.96) | 3.23(2.46) | 1 |
| 2-branch (all normalized) | 22.35(2.34) | 19.02(1.98) | 1 |
| **2-branch** | **3.17 (1.96)** | **2.42 (1.90)** | 1 |
| 2-branch (raw input) | 30.30(4.02) | 28.79(4.71) | 0.01 |
| 2-branch | 3.17 (1.96) | 2.42 (1.90) | 0.01 |
| 2-branch (raw input) | 716.54(438.76) | 797.02(517.23) | 100 |
| 2-branch | 3.17 (1.96) | 2.42 (1.90) | 100 |

Table 2: Mean $T1rho$ prediction value per subject under free breathing scenario of different models (in **ms**). P stands for patient and H stands for healthy subject.

| Subject NO | Gender | 2-TSL | 4-TSL | DL |
|---|---|---|---|---|
| 1 (P) | M | 2235.59 | 63.35 | 44.43 |
| 2 (H) | M | 190.30 | 46.61 | 44.77 |
| 3 (H) | M | 2072.54 | 38.24 | 42.88 |
| 4 (H) | F | 3062.57 | 51.36 | 44.65 |
| 5 (H) | F | 50.26 | 45.83 | 46.66 |
| 6 (H) | F | 3754.38 | 71.17 | 47.26 |

Table 3: Session-wise Wilcoxon Rank Sum testing for $T1rho$ value

| | 2-TSL | 4-TSL | DL |
|---|---|---|---|
| $p$ value | 0.006 | 0.15 | 0.85 |

The result of each subject is shown in Table 2 and visualisation examples are shown in Figure 3. The method of 2-TSL fitting was very sensitive to motion and can produce extremely erroneous value. Erroneous values were also observed in Subject-1 and Subject-6 using the 4-TSL fitting method. The proposed deep learning method produced reasonable values for all the subjects. Both 2-TSL and 4-TSL fitted results had erroneous appearance due to motion. The results of hypothesis testing is shown in Table 3. It can be seen that 2-TSL rejected the null hypothesis, and our method had a much higher confidence for not rejecting the null hypothesis that session-wise mean prediction had the same distribution as the 4-TSL fitting value in breath-hold scenario.

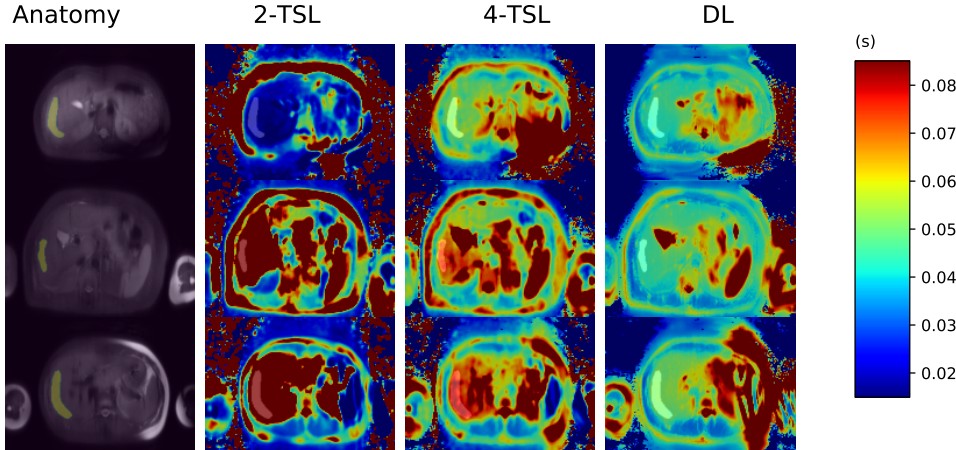

Figure 3: Visualisation of the $T1rho$ maps of different methods. The bright areas indicate the region of interests.

## 4. Limitations

In this work, we do not explicitly decouple the learnt representations of $S_0$ and $T1rho$, while decoupling is supposed to reflect the physics intuition. Also, the scaling model in our method is simplified to uniform scaling. The scaling becomes not uniform over the image if the local tissues are significantly different. Besides, our training pipeline requires the training data to be collected with breath hold. These problems need further exploration in the future.

## 5. Conclusion

We present a preliminary study on inferring $T1rho$ quantification map using a self-supervised physics-informed model, and demonstrates the potential of using only one $T1rho$ weighted image to predict $T1rho$ with applications in quantitative liver T1rho imaging with free-breathing. With the proposed two branch training mechanism, the model generality of signal scale variation is also demonstrated. Future works includes tackling the generality on real data under different scanning protocol settings, MR pulse sequence and machines as well as investigation on clinical application.

## Acknowledgments

This study was supported by a grant from the Innovation and Technology Commission of the Hong Kong SAR (Project MRP/046/20X), a Faculty Innovation Award from the Chinese University of Hong Kong, and a grant from the Research Grants Council of the Hong Kong SAR (Project SEG CUHK02). We would also like to thank Professor Thierry Blu for providing the Matlab code of non-least square fitting.

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

## Appendix A. Explanation on planning during scan session

Before doing the $T1rho$ scan, the planning of FOV and shim box is carried out on frontal and transverse anatomical pre-scans, which is shown in Figure 4. The planning is then applied to the following $T1rho$ scan in both breath-hold and free-breathing scenarios. When doing breath-hold scan in $T1rho$, the actual slice position is different from but near the planned position due to out of plan motion, since the exact breath-hold position cannot be directly controlled by the subject. As for the free breathing scan, the MR machine capture slices near the planned position. The above reason drive us to use a paired distribution hypothesis testing.

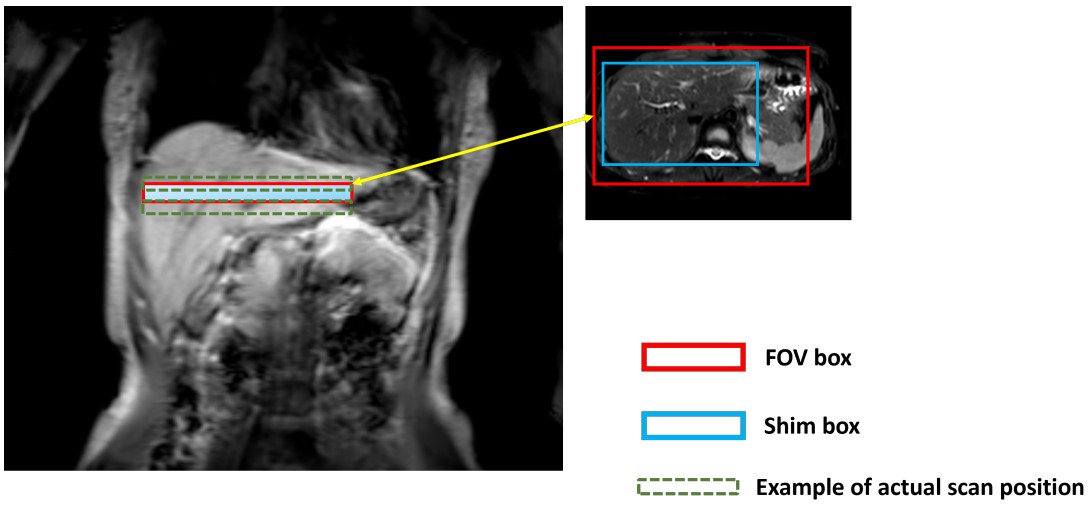

Figure 4: Illustration of planning before $T1rho$ scan

