# OpenReview forum: "Breathing Freely: Self-supervised Liver T1rho Mapping from A Single T1rho-weighted Image"
_MIDL.io/2022/Conference — MIDL 2022_

### Official Review · Reviewer_tfoQ · 2022-01-23

**Confidence:** 4
**Preliminary Rating:** 3
**Recommendation:** Poster

**Summary:**

this paper proposed a physics-informed deep learning model to implicitly decouple the learned representations of S0 and T1rho to quantify T1rho relaxation time.
the authors claimed that the method can achieve a better mapping than the traditional fitting method in free-breathing scenarios.
the model was trained on healthy subjects and evaluated on both healthy subjects and patients.


**Strengths:**

1) a physics-informed deep learning model was designed to implicitly decouple the learned representations.
2) the model was trained on healthy subjects and evaluated on both healthy subjects and patients.
3) the model was evaluated on both breath-hold and breath-free scenarios.

**Weaknesses:**


Technical setting:
1) decoupling the learned representations is great but there is no regularization on the representation especially the target T1rho. One potential solution could be matching the representations of the learned T1rho and referenced T1rho with a discriminator or KL divergence.

Evaluation:
1) Metrics: the right lobe of the liver (as an ROI) is used during the evaluation stage. However, it seems not used in the free breathing scenario. And why use the right lobe? To my understanding, for patients, the ROI should be the pathology.
2) Results: in Section 3.3, the authors mentioned "The mean prediction for every subjects is computed and a pair t-test were carried out to test if the subject-wise mean T1rho is significantly different from the reference". Again, the ROI is not used here. The results would be affected by the noise which is agreed by the authors. How would the results be convincing? In Figure 3, there is a significant difference between the reference and DL-generated T1rho, not only in the lobe area but also in other organs.
3) Statistics test:  In Table 3, how are the p-values calculated? i.e., what are pairs in the test given there are no ground truths?

Presentation:
The presentation of Section 2.2.1 could be improved especially the idea of decoupling.

Typos and format:
Please check the whole manuscript.

For example,
x. 'every subjects'
x. 'pair t-test'
x. there should be a space between the text and the citation.



**Deanonymize Review:**

no

**Detailed Comments:**

(Copy and Paste from previous sections)

Generally, the paper is interesting.
The authors proposed a physics-informed deep learning model to implicitly decouple the learned representations of S0 and T1rho to quantify T1rho relaxation time. The authors claimed that the method can achieve a better mapping than the traditional fitting method in free-breathing scenarios. The model was trained on healthy subjects and evaluated on both healthy subjects and patients.

Technical setting:
1) decoupling the learned representations is great but there is no regularization on the representation especially the target T1rho. One potential solution could be matching the representations of the learned T1rho and referenced T1rho with a discriminator or KL divergence.

Evaluation:
1) Metrics: the right lobe of the liver (as an ROI) is used during the evaluation stage. However, it seems not used in the breath-free scenario. And why use the right lobe? To my understanding, for patients, the ROI should be the pathology.
2) Results: in Section 3.3, the authors mentioned "The mean prediction for every subjects is computed and a pair t-test were carried out to test if the subject-wise mean T1rho is significantly different from the reference". Again, the ROI is not used here. The results would be affected by the noise which is agreed by the authors. How would the results be convincing? In Figure 3, there is a significant difference between the reference and DL-generated T1rho, not only in the lobe area but also in other organs.
3) Statistics test:  In Table 3, how are the p-values calculated? i.e., what are pairs in the test given there are no ground truths?

Presentation:
The presentation of Section 2.2.1 could be improved especially the idea of decoupling.

Typos and format:
Please check the whole manuscript.

For example,
x. 'every subjects'
x. 'pair t-test'
x. there should be a space between the text and the citation.



**Final Rating After The Rebuttal:**

4: Weak Accept

**Justification Of The Final Rating:**

1. A physics-informed model to disentangle two components.
2. Good application for a new dataset (however, the novelty of the application is not within my expertise).
3. Part of the responses address my concerns.

**Paper Type:**

both

**Questions To Address In The Rebuttal:**

1) My comments on the technical setting of the framework.
2) Potential issues in the evaluation setting (e.g. the motivation of ROI and the inconsistent usage in the evaluation), results and statistics test.
3 Presentation, typos and format.

**Special Issue:**

no

---

### Official Review · Reviewer_9cjV · 2022-01-24

**Confidence:** 3
**Preliminary Rating:** 2

**Summary:**

* A self-supervised data consistency loss is proposed for training a CNN that takes a $T1pho$-weighted image as input and outputs
    * a constant $S_0$ map and
    * a $T1pho$ tissue map.

* The CNN is trained on $15$ healthy subjects, and evaluated on $4$ test sets:
    * $4$ healthy subjects (breath-hold),
    * $40$ diseased test subjects (breath-hold),
    * $5$ healthy subjects (free-breathing).
    * $1$ diseased subject (apparent motion).

**Strengths:**

* The manuscript tackles an important problem - that of obtaining $T1pho$ maps from a single $T1pho$-weighted image.
* The proposed self-supervised loss is well-designed to leverage knowledge of the underlying physical model.

**Weaknesses:**

* Some aspects about the experimental setup are unclear to me:
    * What is the point of showing the 2-branch (raw input) evaluations in Table 1? Isn't it quite natural that the lack of normalization causes the network to perform poorly when the input scales are very different than training?
    * It would actually be more interesting to see the performance of a CNN where both branches receive normalized inputs.

* The description of the free-breathing experiment is unclear, even after reading the information in the Appendix. As this experiment is one of the two experiments in this validation paper, I believe it is extremely important to explain this clearly and include the description in the main paper.

**Deanonymize Review:**

no

**Detailed Comments:**

* The general writing of the paper needs to be heavily improved. Please get the paper proof-read by a native English speaker. Several grammatical and punctuation mistakes need to be removed. Some examples
    * There should be no space before a full stop or a comma.
    * There should be a space after a full stop or a comma.
    * These problems needs  $\rightarrow$ These problems need
    * indicating a descent repeatability $\rightarrow$ indicating a decent repeatability


**Final Rating After The Rebuttal:**

3: Borderline

**Justification Of The Final Rating:**

I have increased the rating in light of some of the clarifications. In particular, I recognize that the free breathing experiments may have been unclear to me due to my lack of familiarity with the MR quantification literature. However, even with the rebuttal, I am still unclear how the proposed 2-branch architecture is beneficial as compared to previous work. Further, it also seems to be that the robustness to motion may have been overly-emphasized in the paper, if the motion artifacts are barely present in the first place. Finally, the grammatical mistakes in the paper have not been corrected even after the rebuttal.

**Paper Type:**

validation/application paper

**Questions To Address In The Rebuttal:**

* It is unclear to me, why the network trained on breath-hold images should generalize to free-breathing images. Could this be due to the procedure used for drawing the ROIs, that avoided regions with motion artifacts? If this is the case, is it still reasonable to claim that the method generalizes to free-breathing scenarios?

* From a methodological point of view, the manuscript follows a similar strategy as in several previous works: Liu et al 2021, Varadarajan et al 2021, Wu et al 2021, etc. I believe this is fine for a validation paper. Nevertheless, I think it is important to discuss the difference in design choices with respect to earlier works, and the motivation for these differences. Can the authors kindly provide a discussion about this?
    * For instance: In this manuscript, a 2-branch CNN is used, with one branch predicting $S_0$, and another predicting $T1pho$ tissue map. This is in contrast with the approach in, for e.g., Varadarajan et al 2021, where the $PD$ map plays a similar multiplicative role in the physical model as the $S_0$ here. Can the authors comment on this difference?

**Special Issue:**

no

---

### Meta-Review · Area_Chair_YFAQ · 2022-02-18

**Recommendation:** Accept (Poster)
**Confidence:** 3

**Metareview:**

This work is not at the core of my expertise, however, the authors could address some of the reviewers concerns. The work seems interesting, the application relatively novel and the evaluation sound. However, some aspects of the method are not completely clear. My final vote is a weak accept, however, the work would certainly benefit from another round of revision. The authors should correct the remaining grammar mistakes.

---

### Decision · Program_Chairs · 2022-02-28

Accept